# Descriptive Epidemiology of Hospitalization of Patients with a Rare Tumor in an Italian Region

Alessandra Rosa [1,*], Vincenzo Fontana [1], Rosa Angela Filiberti [1], Paolo Pronzato [2] and Matilde Mannucci [1]

[1] Clinical Epidemiology Unit, IRCCS Ospedale Policlinico San Martino, 16132 Genoa, Italy
[2] Medical Oncology Unit, IRCCS Ospedale Policlinico San Martino, 16132 Genoa, Italy
* Correspondence: alessandra.rosa@hsanmartino.it

**Abstract:** Objectives: Rare tumors (RT) collectively account for one quarter of all malignancies in Italy. The low frequency and the large heterogeneity in natural history and outcome of individual diseases, together with a scarcity of epidemiological information make them a challenge for clinical practice, as well as for public healthcare organizations. We conducted a retrospective study to quantify the burden of hospitalization in a real-word setting in patients diagnosed with these diseases in an Italian region. Methods: RT patients were tracked along all hospital stays from 2000 to 2019 using hospital discharge records. Frequency of hospitalizations, average time spent in hospital and median timespan between consecutive admissions were considered. Re-hospitalization rates were analyzed through a multivariable negative binomial regression analysis to adjust for confounding and allowing for over-dispersion in count data. Results: As a whole, 57,329 patients were identified at first stay for all studied tumors. A total of 183,959 admissions were retrieved, along a median of 3 hospitalizations per patient. Median timespan between hospitalizations shortened in the course of the study years (12.5 months in 2000–2004 to 5.4 months in 2015–2019). The overall re-hospitalization rate increased from 0.92 per patient/year (95% CI = 0.81–1.04) in 2000–2004 to 2.17 (95% CI = 1.90–2.47) in 2015–2019. Conclusions: Overall, the hospitalization rate of patients with a RT increased in the twenty years since the 2000 and particularly doubled starting from 2015. A higher burden of hospitalizations was found for tumors of the central nervous system, thoracic cavity, digestive tract and sarcomas. To the best of our knowledge this is the first paper related to access to Italian healthcare facilities of patients with these tumors.

**Keywords:** rare tumors; hospitalization; re-hospitalization; healthcare





## 1. Introduction

Rare Tumors (RT) are a heterogeneous group of diseases collectively accounting for one quarter of all malignancies in Italy. The low frequency and the large heterogeneity in natural history, biological behavior and outcome of individual diseases, together with a scarcity of epidemiological information, make them a challenge for clinical practice, as well as for public healthcare organizations. Major problems are inappropriate diagnosis, lack of therapies and of evidence-based treatment guidelines, as well as difficulties in conducting clinical trials [1–4]. So far, knowledge on the burden of these pathologies on society is limited [5–7].

The first issue is the number of diseases that are included under the umbrella term of "rare cancers". Several lists have been proposed by different institutions according to tumor prevalence [8,9]. The Surveillance of Rare Cancers in Europe (RARECARE) project, by using an annual incidence rate of less than 6 per 100,000 inhabitants, has estimated that these tumors account annually for up to 22% of all new cancers [1] and about 4% of total cancer prevalence in Europe [2,10].

Clinical management of RT often requires a multidisciplinary approach. In the last years, diagnostic and therapeutic pathways of patients with RT have changed due to

technological and organizational innovations, with an increase of access to health services. Hospitalizations are likely to be increased, too, but there is limited research addressing this topic [10]. To the best of our knowledge there are no published papers related to access to Italian healthcare facilities of patients with these tumors.

Awareness of the frequency and reasons of hospitalizations is a fairly effective way to understand the healthcare burden and the needs for care. Moreover, hospitalizations may be used as an indicator of the performance of the healthcare system and may help in taking actions to enhance the clinical management of patients with these neoplasms [1].

We conducted a prospective non-concurrent study to quantify and describe the burden of hospitalization in a real-word setting in patients diagnosed with RT in an Italian region. Frequency, patterns of readmission and timing of hospitalizations of patients were explored.

## 2. Material and Methods

### 2.1. Study Patients

The data source used in this study were the hospital discharge records (HDR) of the Liguria region (LR), which contain information on all patients discharged from private or public hospitals after a planned or urgent (diagnostic or interventional) admission. In Italy, healthcare is covered almost entirely by the National Health Service (NHS) and the use of this database makes it possible to track virtually all the hospitalizations. HDR were primarily intended for administrative aims to obtain refunds, but are being used also for epidemiological and public health purposes. LR is one of the 20 administrative regions of Italy, located in the north-west of the country, with a population of approximately 1.5 million inhabitants.

The database records demographic data, codes of the hospital and department of admission, type of hospitalization (ordinary or day hospital), admission and discharge date and clinical procedures. The primary diagnosis at discharge and up to seven secondary comorbidities are reported. Diagnoses are codified at each hospital according to the Italian version of the 9th International Classification of Diseases, Ninth Revision, Clinical Modification (ICD-9-CM) [11].

Patients included in this study had been hospitalized in LR with at least one primary diagnosis of RT between 1 January 2000 and 31 December 2019.

We enrolled the patients starting from 2000 due to the low accuracy and comparability of SDO diagnostic codes previously. The discharge data of patients resident in LR but hospitalized in other Italian regions were also collected.

Based on selection criteria from the Italian Rare Cancer Network (IRCN), the following groups of cancers were considered: epithelial tumors of the head and neck (HNT: tongue, major salivary glands, gum, mouth, oro- and naso-pharynx); epithelial tumors of the thoracic cavity (TCT: trachea, pleural mesothelioma, thymus); epithelial tumors of the digestive system (DGT: small intestine, anus, liver, gallbladder, bile ducts); endocrine glands (EGT: including thyroid); epithelial tumors of the urinary system and tumors of male genital system (GUT: testis, penis and other male genital organs; renal pelvis, ureter and urethra); tumors of the female genital system (FGT: vagina, clitoris and vulva) and of the central nervous system (CNS: eye, brain, other and unspecified parts of nervous system); sarcomas (SAR: bone and articular cartilage, connective and soft tissue); rare hematological diseases (HET: multiple myeloma and immunoproliferative neoplasms, leukemias); and epithelial skin tumors (SKT: skin of trunk except scrotum, labia) (Table S1: Selection of patients with rare tumors according to ICD-9-CM classification in discharge diagnosis). All patients were identified through the selected ICD (International Classification of Diseases) codes in all diagnostic fields and tracked along all hospital stays over the study period using a unique identification code for each Italian resident (fiscal code). The index hospitalization was the first admission ($H_0$) with the above mentioned discharge diagnoses in the period of interest.

Patients were followed up until the end of 2019. Vital status was documented through the Liguria Mortality Registry, municipal demographic databases and HDR discharge status.

This study was approved by the Ethics Committee of LR and was conducted in compliance with the principle of the Declaration of Helsinki. Patients were not directly involved. HDR are recorded with the patient's consent and can be used as aggregated data for scientific studies without further authorizations.

### 2.2. Study Outcomes

The outcome of the study was to characterize the hospitalization history of patients with RT using admissions at $H_0$, total number of admissions and readmission rates for all RT-related hospitalizations in and out LR. The average time spent in hospital and the timespan between consecutive admissions were also calculated to evaluate the longitudinal trend of hospitalization.

### 2.3. Statistical Analysis

HDR data were analyzed according to patients' baseline main characteristics, namely gender, age at $H_0$ (split into quartiles: 0–56, 57–69, 70–78, 79–103), period of $H_0$ (split into five-year categories: 2000–2004, 2005–2009, 2010–2014, 2015–2019) and vital status at the end of follow-up.

The healthcare burden represented by multiple admissions of patients with RT was firstly evaluated by computing the median time between two consecutive admissions. In order to avoid excessive random variability in median estimates due to a small sample size, a maximum of 15 readmissions per patient were considered for analyses. In practice, about 4% (7194 out of 183,959) of all hospitalizations were excluded. Secondly, a multivariable negative binomial regression analysis was applied to re-hospitalization rates in order to evaluate the joint effect of patients' baseline characteristics while adjusting regression estimates for confounding and allowing for overdispersion in count data [12].

All statistical indexes were accompanied by 95% confidence intervals (95% CI) or inter-quartile range (IQR) as measures of sampling variation. A *p*-value less than or equal to 0.05 was considered as statistically significant.

All statistical analyses were carried out using Stata software [13].

## 3. Results

In the years 2000–2019 we identified 57,329 patients (56.3% males; 30.4%, 26.0%, 22.5% and 21.0% in each calendar period at $H_0$) with 183,959 hospitalizations, of which 158,490 (86.2%) were re-hospitalizations for all studied tumors (Table 1). Twenty-five percent of hospitalizations were related to patients younger than 56 years, while 50% concerned patients aged 70 years or older. Four percent of all cases were younger than 20 years and were mainly affected by SAR and CNS (11%), EGT (8%) and HET (5%) (Table S2: Distribution of age at $H_0$ by group of rare tumors).

About 56% of patients were re-hospitalized at least once and about 20% of them had up to five stays, while another 2% had 15 or more hospitalizations (Table 1). First hospitalizations were higher for HET (22.2%), DGT (20.4%) and CNS (13.6%) patients. The lowest numbers were found for FGT (vagina, clitoris, vulva: 1.7%; skin of labia: 0.1%) and SAR (5.1%).

Overall, the percentage of re-hospitalization tended to decrease from 2000–2004 to 2015–2019 (from 36% to 14%) (Table 1).

At least one re-hospitalization was registered among approximately 70% of patients with HET or TCT, 60% of those with HNT or DGT, 50% of patients with SAR, CNS or FGT and among 40% of patients with EGT or GUT (Table S3: Distribution of re-hospitalization by group of rare tumors and calendar period).

**Table 1.** Characteristics of patients with rare tumors in Liguria Region during 2000–2019.

| Characteristics & Categories | Patients at $H_0$ [a] | | Re-Hospitalization | |
|---|---|---|---|---|
| | N [b] | % [c] | N [b] | % [c] |
| Tumor Groups | | | | |
| Sarcomas (SAR) | 2932 | 5.1 | 9232 | 5.8 |
| Central Nervous System (CNS) | 7789 | 13.6 | 19,489 | 12.3 |
| Head and Neck (HNT) | 4482 | 7.8 | 10,404 | 6.6 |
| Hematological (HET) | 12,747 | 22.2 | 61,835 | 39.0 |
| Thoracic Cavity (TCT) | 3458 | 6.0 | 9985 | 6.3 |
| Genitourinary System (GUT) | 3713 | 6.5 | 4912 | 3.1 |
| Female Genital System (FGT) | 967 | 1.7 | 1621 | 1.0 |
| Digestive System (DGT) | 11,710 | 20.4 | 29,855 | 18.8 |
| Endocrine Glands (EGT) | 5242 | 9.1 | 8981 | 5.7 |
| Skin (SKT) | 4289 | 7.5 | 2176 | 1.4 |
| Skin of Trunk [d] | 4207 | 7.3 | 2031 | 1.3 |
| Skin of Labia [e] | 82 | 0.2 | 145 | 0.1 |
| Gender | | | | |
| Male | 32,259 | 56.3 | 95,525 | 60.3 |
| Female | 25,070 | 43.7 | 62,965 | 39.7 |
| Age at $H_0$ [a] | | | | |
| 0–56 | 14,332 | 25.0 | 54,188 | 34.2 |
| 57–69 | 14,333 | 25.0 | 44,682 | 28.2 |
| 70–78 | 14,331 | 25.0 | 37,276 | 23.5 |
| 79–103 | 14,332 | 25.0 | 22,344 | 14.1 |
| Period of $H_0$ [a] | | | | |
| 2000–2004 | 17,439 | 30.4 | 57,306 | 36.2 |
| 2005–2009 | 14,923 | 26.0 | 44,655 | 28.2 |
| 2010–2014 | 12,917 | 22.5 | 33,624 | 21.2 |
| 2015–2019 | 12,050 | 21.0 | 22,905 | 14.5 |
| Vital Status [f] | | | | |
| Alive | 26,709 | 46.6 | 61,304 | 38.7 |
| Died | 30,620 | 53.4 | 97,186 | 61.3 |
| Total | 57,329 | 100.0 | 158,490 | 100.0 |

[a] index hospitalization (first admission); [b] absolute frequency; [c] percent frequency; [d] ICD9 code 173.5, skin of trunk except scrotum; [e] ICD9 code 184.1, 184.2, skin of labia minora and majora; [f] vital status at last discharge.

In the whole sample, the medians of the number of hospitalization and days of hospital stay were 2 (IQR = 1–4) and 16 (IQR = 5–42), respectively (Table 2). Length of stay was longer for patients with HET (41 days, IQR = 14–94), TCT (25 days, IQR = 12–46), DGT (22 days, IQR = 10–41) and HNT (18 days, IQR = 6–40). The lowest median was observed in patients with EGT (4 days, IQR = 2–10) and SKT (2 days, IQR = 1–4). Days of stay decreased from a median of 19 (IQR = 6–48) in 2000–2004 to 13 (IQR = 3–33) in 2015–2019 (Table 2).

Overall, the median timespan between hospitalizations was 9.5 months (IQR = 2.7–28.0), characterized by a reduction in patients >78 years (6.2 months, IQR = 1.9–18.5) and a decreasing trend over the study period (12.5 months, IQR = 3.5–38.9 in 2000–2004 to 5.4 months, IQR = 1.9–13.2 in 2015–2019) (Table 2, Figure 1). A lower median timespan was observed for patients with GUT (4 months, IQR = 1.5–17.5) (Table 2, Figures S1–S10: Median timespan and 95% confidence interval between two consecutive hospitalizations for each rare tumor in the Liguria region, 2000–2019).

About 10% of all hospitalizations were in a pediatric hospital. Forty-one percent of $H_0$ involved medical wards and thirty-three percent, surgery departments. Thirteen percent of patients were hospitalized in an oncology ward. This percentage increased in the following hospitalization up to 54% for all patients and to 80% for patients with the first stay from 2015. About 36% of all hospitalizations were planned, while around 26% (32% at $H_0$) were due to emergency situations (Table S4: Type of hospitalization of patients with rare tumors in Liguria Region during 2000–2019).

**Table 2.** Hospitalization characteristics of patients with rare tumors in Liguria Region during 2000–2019.

| Characteristics & Categories | Hospitalizations | | Days of Stay | | Months between Hospitalizations | |
|---|---|---|---|---|---|---|
| | Median | IQR [a] | Median | IQR [a] | Median | IQR [a] |
| Tumor Groups | | | | | | |
| Sarcomas (SAR) | 3 | 1–7 | 14 | 4–40 | 9.0 | 3.4–27.9 |
| Central Nervous System (CNS) | 2 | 1–4 | 15 | 4–37 | 7.9 | 2.1–22.3 |
| Head and Neck (HNT) | 2 | 1–4 | 18 | 6–40 | 7.7 | 2.0–23.8 |
| Hematological (HET) | 4 | 2–8 | 41 | 14–94 | 14.2 | 4.5–40.0 |
| Thoracic Cavity (TCT) | 2 | 1–4 | 25 | 12–46 | 8.4 | 3.0–17.3 |
| Genitourinary System (GUT) | 1 | 1–2 | 10 | 4–20 | 4.0 | 1.5–17.5 |
| Female Genital System (FGT) | 2 | 1–3 | 13 | 5–25 | 9.9 | 2.5–27.2 |
| Digestive System (DGT) | 2 | 1–4 | 22 | 10–41 | 7.6 | 2.2–22.4 |
| Endocrine Glands (EGT) | 2 | 1–3 | 4 | 2–10 | 7.7 | 2.8–31.1 |
| Skin (SKT) | 1 | 1–2 | 2 | 1–4 | 29.1 | 7.6–76.0 |
| Gender | | | | | | |
| Male | 2 | 1–5 | 18 | 5–45 | 9.6 | 2.8–27.4 |
| Female | 2 | 1–4 | 14 | 4–38 | 9.4 | 2.7–28.9 |
| Age at $H_0$ [b] | | | | | | |
| 0–56 | 3 | 1–7 | 13 | 3–54 | 10.3 | 3.3–29.9 |
| 57–69 | 3 | 1–5 | 19 | 5–50 | 11.7 | 3.1–35.2 |
| 70–78 | 2 | 1–4 | 19 | 6–43 | 10.2 | 2.8–29.7 |
| 79–103 | 2 | 1–3 | 15 | 6–29 | 6.2 | 1.9–18.5 |
| Period of $H_0$ [b] | | | | | | |
| 2000–2004 | 2 | 1–5 | 19 | 6–48 | 12.5 | 3.5–38.9 |
| 2005–2009 | 2 | 1–5 | 17 | 5–44 | 11.0 | 3.0–33.5 |
| 2010–2014 | 2 | 1–5 | 16 | 4–41 | 9.3 | 2.6–26.6 |
| 2015–2019 | 2 | 1–4 | 13 | 3–33 | 5.4 | 1.9–13.2 |
| Vital Status [c] | | | | | | |
| Alive | 2 | 1–5 | 8 | 3–27 | 8.6 | 2.6–29.4 |
| Died | 2 | 1–5 | 25 | 10–50 | 10.1 | 2.8–27.3 |
| Total | 2 | 1–4 | 16 | 5–42 | 9.5 | 2.7–28.0 |

[a] inter-quartile range; [b] index hospitalization (first admission); [c] vital status at last discharge.

At $H_0$, about 68% of all patients were hospitalized in ordinary regimen and the percentage increased in the study years in all hospitalizations ($H_0$: from 65% of hospitalizations in 2000–2004 to 77% in 2015–2019; fifth re-hospitalization: from 59% in 2000–2004 to 65% in 2015–2019). On the other hand, in the same years, a decreasing percentage was observed at $H_0$ for patients attending a day hospital (Table S5: Regimen of hospitalization of patients with rare tumors in Liguria Region during 2000–2019). Among these patients more than 80% with $H_0$ in 2015–2019 were administered therapies during the first five re-hospitalizations (Table S6: Hospitalization regimen in patients with rare tumors in Liguria Region during 2000–2019 by calendar period and re-hospitalization order, Table S7: Reasons for hospitalization and re-hospitalization in day hospital regimen of patients with rare tumors in Liguria Region during 2000–2019 by readmission order and calendar period).

The results of the negative binomial regression analysis are reported in Table S8 (Figure 2) as re-hospitalization rates and rate ratios (RR), along with the corresponding 95% CI and *p*-value. A slight increase with age (+2%) was found for patients older than 78 (2.17, 95% CI = 1.90–2.47) when compared to patients younger than 57 years (2.13, 95% CI = 1.87–2.43). Fifty-three percent of patients had died and showed a rate which was about 20% higher (RR = 1.21, 95% CI = 1.15–1.28, *p*-value < 0.001) than patients who were still alive by the end of follow-up.

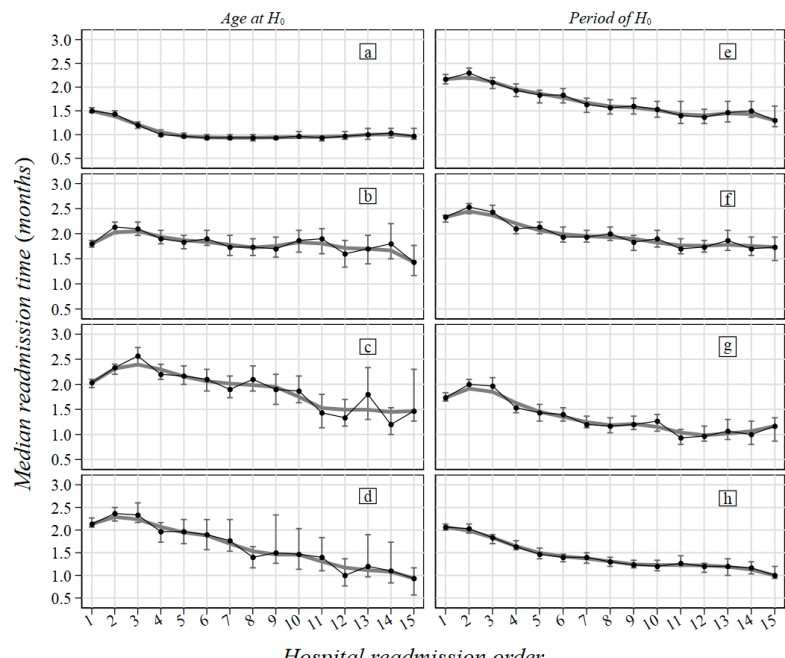

**Figure 1.** Median timespan between two consecutive hospitalizations for rare tumors by age at $H_0$ and period of $H_0$ in Liguria Region, 2000–2019. Legend—$H_0$: index hospitalization (first admission); 95% CI: 95% confidence interval for median timespan; (**a**) 0–56 age group at $H_0$; (**b**) 57–69 age group at $H_0$; (**c**) 70–78 age group at $H_0$; (**d**) 79–103 age group at $H_0$; (**e**) 2000–2004; (**f**) 2005–2009; (**g**) 2010–2014; (**h**) 2015–2019.

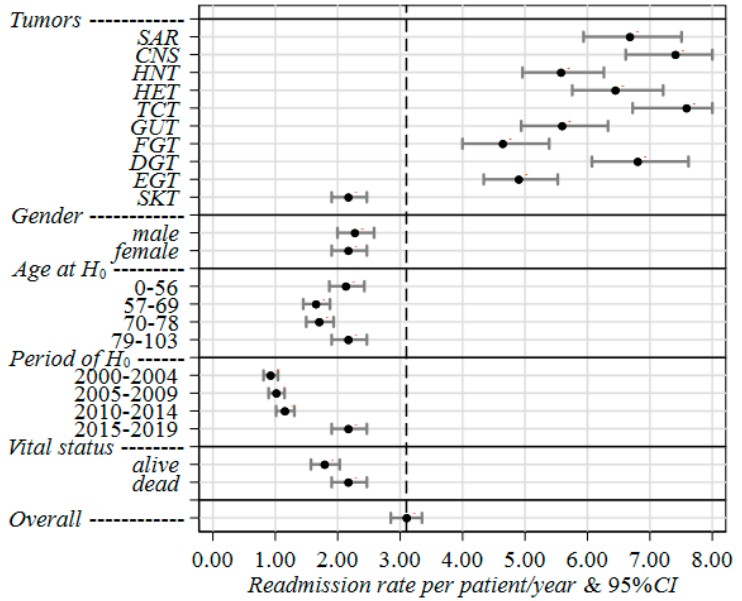

**Figure 2.** Joint effect of tumor group, gender, age at $H_0$, period of $H_0$ and vital status at last discharge on readmission rates of rare tumors patients in Liguria Region during 2000–2019, estimated through the multivariable negative-binomial regression method. Legend—$H_0$: index hospitalization (first admission); Rate: readmission rate per patient/year; 95% CI: 95% confidence interval; SAR: sarcomas; CNS: central nervous system; HNT: epithelial tumors of head and neck; HET: hematological malignancies; TCT: epithelial tumor of thoracic cavity; GUT: urinary system and male genital organs; FGT: female genital system; DGT: epithelial tumor of digestive system; EGT: tumor of endocrine glands; SKT: epithelial tumor of skin.

A noteworthy difference in re-hospitalization rates was pointed out regarding periods of $H_0$, with a statistically significant increase with years of first hospitalization. In particular, the rate of re-hospitalization in 2015–2019 was more than twice (RR = 2.35, 95% CI = 2.18–2.55, *p*-value < 0.001) that in 2000–2004 (Table S8, Figure 2).

A higher burden of hospitalizations was found for TCT (7.58, 95% CI = 6.72–8.55), CNS (7.40, 95% CI = 6.61–8.29), DGT (6.80, 95% CI = 6.07–7.62), SAR (6.68, 95% CI = 5.94–7.51), and HET (6.44, 95% CI = 5.76–7.21). Conversely, low rates were found for SKT (2.17, 95% CI = 1.90–2.47 (Table S8, Figure 2).

## 4. Discussion

We performed an analysis of discharge records to explore the burden of hospitalizations in patients with RT in a period of twenty years in private or public health centers in an Italian region.

First hospitalizations were higher for HET, DGT and CNS tumors. Twenty percent of patients had up to five stays. Length of stay at $H_0$ decreased along the years, but a shorter median timespan between hospitalizations was pointed out especially for patients with GUT.

Overall, the re-hospitalization rate adjusted for confounding increased in a statistically significant manner over the study years and doubled in the last period of first hospitalization (years 2015–2019).

A higher burden of hospitalizations was found for TCT, CNS, DGT, SAR and HET.

RT are not so rare, accounting in Europe for 24% of all incident neoplasms, but the low prevalence of single pathologies makes them a significant problem in patient management [1]. In spite of the pathologic and clinical heterogeneity, RT patients experience similar problems, from diagnosis to treatment, owing to the need of reference centers and a multidisciplinary approach.

To date, RT have been object of few investigations and questions still remain related to their epidemiology, biology and clinical practice. The first issue is the definition of these diseases. RT have been defined as "any tumor that involve a small percentage of the population in any age group, gender or organ". The group includes rare histologic variants and rare cases of common tumors occurring in uncommon hosts such as male breast cancer, but these diseases have been internationally classified according to different criteria based on disease prevalence or incidence [3,4,8,9,14–17]. The European Union through the RARECARE project has proposed a definition based on cancer incidence and generated a list of 198 RT. Evaluating data from ninety-four population-based cancer registries in 21 European countries and combining topography and morphology codes of the ICD for oncology, RT have been defined as those with an incidence rate less than 6 per 100,000 per year [2,17–19].

Based on this definition, the estimated annual incidence rate of all RT in Italy is 147 per 100,000 inhabitants, corresponding to about 89,000 new cases (25% of all incident cancers) and about 900,000 people are alive with one of these diseases. More recently, tumors have been grouped within 12 families of RT; in this way, RT correspond to 10–20% of all cancer cases [20]. Estimates of new RT cases for Italian regions show a large variability between different areas. Due to low prevalence, epidemiologic studies have difficulties in identifying indisputable etiological risk factors, nevertheless it is possible that the differences in incidence are due to diverse environmental or individual conditions as well as to heterogeneity in diagnostic capacity [2].

The mean age of patients with RT is lower than those with common cancers and all childhood cancers can be considered as RT. Hematological and solid tumors account for about 7% and 18% of new RT cases, respectively. When considering prevalence, the highest rates are observed for rare hematological diseases and for RT of the female genital system [1,2]. In our series, 4% of all cases were younger than 20 years; 6% to 12% had HET, EGT, SAR or CNS while 1% had GUT or HNT.

RT are thought to constitute a major public health problem, but their overall burden on society has been addressed by few authors [1]. These tumors can be challenging for patients and clinicians, especially to diagnose. Patients with these pathologies are assumed to use hospitals often, but scarce evidence exists regarding the frequency and reasons of readmissions. A better awareness of the demands of care for these patients and their burden on society is necessary for planning health services and to assess health needs and costs [1].

Due to the low number of patients, it is challenging to conduct rigorous clinical trials and so far there is a scarcity of effective treatments and of sufficient evidence-based treatment guidelines [4]. Nevertheless, over the last few years new models of care have been developed, with a multidisciplinary approach to diagnosis and treatment. Genomic analyses applied to common cancers allowed the identification of aberrations in RT and at the same time these diseases show genomic homogeneity, allowing molecular findings that are pertinent to common cancers. If for many rare cancer types a genomic understanding remains incomplete or absent, a number of them may benefit from molecular target therapy [21]. A very recent systematic review has also evidenced a clinical benefit from treatment with immune checkpoint inhibitors for some conditions [22].

Due to large heterogeneity (tumor site, natural history, molecular characteristics), there is consensus that patients with RT should be treated at centers of reference such as those adhering to EURACAN, the patient-centered multi-domain European Reference Network, bringing together different groups of rare cancers, based on histological and organ of origin classifications (https://euracan.eu/), with application of standard management protocols [10,23]. On the other hand, a drawback of centralization may be the need for patients to move to the center and thus to increase the risk of long waiting lists [24].

At present no tools for early detection are available for rare cancers and yet it is necessary to raise public awareness of these tumors, e.g., sharing information on social media.

*Strengths and Limitations*

To our knowledge, this is the first study on the burden of hospitalizations for a large number of RT and a wide period in Italy.

We think that the study shows an acceptable level of internal validity mainly because we resorted to a standardized health information system (HDR) used not only in clinical settings but also in environmental epidemiology and public health studies. Usually, few sources are available to obtain information on hospitalizations. The regional discharge data allow a systematic and economic way to collect clinical information and to reconstruct the hospitalization history of each resident. Some surveys have supported the use of these data to estimate geographic differences in population prevalence of non-neoplastic diseases [11]. It has also been shown that HDR are a valuable tool in cancer surveillance for identifying incident cases, quantifying the cancer care needs of patients and monitoring geographical and temporal variation of clinical practice in different types of cancer [24–29]. Their use may be a chance to plan health services.

In respect to external validity or generalizability, if we assumed all Italians as the target population, we would draw a valid epidemiological inference given that HDR represent a nationwide, standardized health information system, but on the other hand we would address the issue of heterogeneity among regional healthcare service quality (e.g., south-to-north gradient) which could seriously prevent extending our findings to the target population [30].

Given the very low occurrence of each single tumor, we included all HDR which fulfilled the abovementioned selection criteria. HDR selected for analysis were statistically pre-processed in order to evaluate the amount of missing data and, wherever possible, to replace them with valid data. We think that this choice allowed us to obtain a satisfying research quality by increasing the validity and precision, with a reasonably low uncertainty in parameter estimates (95% CL) due to the sampling variability.

Nevertheless, there are some limitations. HDR were primarily intended for administrative purpose to obtain refunds and may generate misclassifications in health services research and give misleading information on patients' health status.

The major concern arises from the ICD-9-CM code used to identify the diseases of interest: the coding system may be quite imprecise for describing specific conditions and may have gone through significant changes in the study years. It is also possible that different ways of coding among physicians may lead to misclassification of some neoplasms. As concerns our topic, RT should be identified by a combination of morphologic and topographic codes that are not available with ICD-9-CM code. For this reason, we used the classification based on the IRCN code which encompasses patients diagnosed with RT. Nevertheless, an overestimation of cancer cases may have occurred with the inclusion of other more commons tumors.

Another drawback is that HDR do not permit to estimate the total burden of a disease since only treatments performed in hospital are recorded and there is no information on outpatient therapies.

As far as reproducibility is concerned, in the scientific literature we did not find a significant amount of similar works. Although this confirms the originality of our analysis, it definitely impedes any, albeit informal, meta-comparisons [31].

## 5. Conclusions

This study revealed that the hospitalization rate of patients with RT increased in twenty years from 2000 and doubled starting from 2015. Compared with a shorter length of stay, patients were admitted to hospital with a higher frequency. A higher burden of hospitalizations was found for TCT, CNS, SAR and DGT. In the last years technological and organizational innovations have improved healthcare of patients, nevertheless RT are still a challenge for patients and clinicians due to various reasons, from lack of scientific knowledge, to the possibility of misdiagnosis and delay in diagnosis, to lack of appropriate therapies. While waiting for a tool for early detection, knowledge of the needs of patients and of the burden of hospitalizations may help allocate resources and improve healthcare quality.

**Supplementary Materials:** The following supporting information can be downloaded at: https://www.mdpi.com/article/10.3390/curroncol29120762/s1, Figures S1–S10: Median timespan and 95% confidence interval between two consecutive hospitalizations for each rare tumor in the Liguria region, 2000–2019; Table S1: Selection of patients with rare tumors according to ICD-9-CM classification in discharge diagnosis; Table S2: Distribution of age at $H_0$ by group of rare tumors; Table S3: Distribution of re-hospitalization by group of rare tumors and calendar period; Table S4: Type of hospitalization of patients with rare tumors in Liguria Region during 2000–2019; Table S5: Regimen of hospitalization of patients with rare tumors in Liguria Region during 2000–2019; Table S6: Hospitalization regimen in patients with rare tumors in Liguria Region during 2000–2019 by calendar period and re-hospitalization order; Table S7: Reasons for hospitalization and re-hospitalization in day hospital regimen of patients with rare tumors in Liguria Region during 2000–2019 by readmission order and calendar period; Table S8: Joint effect of tumor group, gender, age at H0, period of H0 and vital status at last discharge on re-hospitalization rates of patients with rare tumors in Liguria Region during 2000–2019, estimated through the multivariable negative-binomial regression method.

**Author Contributions:** Conceptualization, V.F. and R.A.F.; methodology, V.F. and R.A.F.; formal analysis, A.R., V.F. and M.M.; investigation, P.P.; data curation, A.R. and M.M.; writing—original draft, A.R., V.F., R.A.F., P.P. and M.M.; writing—review & editing, A.R., V.F., R.A.F., P.P. and M.M.; funding acquisition, V.F. All authors have read and agreed to the published version of the manuscript.

**Funding:** This work was partially funded by the Italian Ministry of Health, RC 2022, Project code R754A, IRCCS Ospedale Policlinico, San Martino, to Vincenzo Fontana.

**Institutional Review Board Statement:** The study was conducted in accordance with the Declaration of Helsinki, and approved by the Ethics Committee of Liguria Region (CER Liguria 630/2021, ID 11928).

**Informed Consent Statement:** Informed consent was obtained from all the participants.

**Data Availability Statement:** The datasets generated and analyzed during the current study are available from the corresponding author on reasonable request. All statistical analyses were carried out using Stata software (StataCorp. Stata: Release 17. Statistical Software. College Station, TX. 2021).

**Conflicts of Interest:** The authors declare that they have no conflict of interest.

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
