# Peer review of "Descriptive Epidemiology of Hospitalization of Patients with a Rare Tumor in an Italian Region"

_curroncol, doi:10.3390/curroncol29120762_

Round 1

Reviewer 1 Report

In the paper"Descriptive epidemiology of patients with a rare tumor in an Italian Region," the authors present changes in the care of patients with rare tumors and challenges both for clinicians and the healthcare system. With the growing costs of contemporary medical treatment, these aspects are of marked importance.

As far as the manuscript is concerned, I have not found Tables S5, 7, S1 (there are only tables 1-3) and Figures S1 to S10 (there are figures 1 and 2).

Please, correct the numbers of tables and figures in the text. 

Reviewer 2 Report

Thank you very much for the opportunity to review this interesting paper. The topic is of major interest in current oncology research: the burden of rare tumors (RT) from the hospital-based perspective. The interesting feature of the paper is that it measures this burden from a hospital-based setting and provides indicators that are very useful and applied in other studies in the same setting.

The authors quantify the burden of hospitalization using real-word data from patients diagnosed with RT in the Liguria Region in Italy. Lost to follow-up of patients is “controlled” as authors state that “the discharge data of patients resident in that Italian region but hospitalized in other regions were also collected.” Methods are described correctly and are appropriate for the analysis. In the results section, I found of interest Figure 1, which describes the time trend of the “Median timespan between two consecutive hospitalizations” according to age and period at first admission.  The main result that must be and advice for managing the future burden of RC is that “the hospitalization rate of patients with RT increased in twenty years from the 2000s and doubled starting from 2015s”.

There are two minor points:

1.- Information of Figure 2 depicts the numbers presented in Table 3, but it is repeated information. I suggest keeping Figure 2 in the main text, since it is very informative and a “fast track” to assess the results, but move Table 3 to supplementary material.

2.- Despite the data comes from hospital-based registry, is there any way to quantify the incidence (incidence rates) in that region of Italy during the study period? If so, authors could provide “the expected time (T) in years needed to observe one new case in the region” as shown in Botta et al 2018 .

3.- Based on the response to point 2, authors must note in the discussion that between region, and even within-region, variability is of interest for public health policies. In this line, etiological research on rare tumors with putative risk factors is called for. (See Salmerón et al 2021).

References:

R1.- Botta L, Capocaccia R, Trama A, Herrmann C, Salmerón D, De Angelis R, Mallone S, Bidoli E, Marcos-Gragera R, Dudek-Godeau D, Gatta G, Cleries R; RARECAREnet Working group. Bayesian estimates of the incidence of rare cancers in Europe. Cancer Epidemiol. 2018 Jun;54:95-100.

R2.- Diego Salmerón, Laura Botta, José Miguel Martínez, Annalisa Trama, Gemma Gatta, Josep M Borràs, Riccardo Capocaccia, Ramon Clèries, for the Information Network on Rare Cancers (RARECARENet) Working Group, Estimating Country-Specific Incidence Rates of Rare Cancers: Comparative Performance Analysis of Modeling Approaches Using European Cancer Registry Data, American Journal of Epidemiology, Volume 191, Issue 3, March 2022, Pages 487–498,

Reviewer 3 Report

The authors of the current research conducted a prospective, non-concurrent study to assess and describe the burden of hospitalization for patients with RT in an Italian Region. Overall, this manuscript clearly lacks of novelty and findings need further, more methodical research. The following is a list of both major and minor concerns regarding the article.

1)      Abstract should be improved.

2)      The source (specific age) and study (only hospital discharge records) of the population is very limited.

3)      Did the team obtain permission for the use of questionnaires from the original author? If yes, please mention this in the text.

4)      The study limitations have been discussed completely; however, the strengths of this study should be mentioned.

5)      What standards did this survey use for inclusion and exclusion?

6)      “A p-value below 0.05 is considered statistically significant”. Please mention them.

7)      How can we raise public awareness for rare tumors (RT)? Should be added in discussion part.

8)      Why other Barometer surveys were not used/designed to know about the respondents awareness regarding RT.

9)      Which types of medicines and technologies could help in diagnosis and prevention of rare tumors? Include it.

10)  Does the sample size match the quality criteria for survey research and reports? Add information.

11)  All acronyms have to be spelled out the first time they are used.

12)  Limitations (the final paragraph of the discussion section) should have their own heading and be placed after the conclusion heading or add subsections “Limitations” in the discussion section to make limitations easier to find in the whole text.

13)  I think the conclusion should need refining.

Round 2

Reviewer 3 Report

The authors should work again on the following questions -

1)      The abstract section should be updated with an emphasis on the problem statement and novelty of the work

2)      The introduction section requires drastic improvement. It should compile critical findings from the preceding studies and highlight the research gaps that motivate authors to carry on with this study. Need to cite some latest reference regarding this study.

3)      The authors should expand the section on materials and methods.

4)      A sentence,  p-value below 0.05 is considered statistically significant” was added by authors only in Material and Methods section. They didn’t apply and mention it in results/and disccusion.

5)      I am not satisfied with the answer given by the authors of the quality criteria for survey research and reports. Add and provide an answer based on previously established information or literature that the survey met the selection criteria. 

6)      Give a brief discussion about the most common treatment ways.

7)      The reference style should be revised and be in accordance with the journal format.

Round 3

Reviewer 3 Report

 The Authors gave careful consideration to every comment. Along with the earlier, much information was added. The manuscript can be accepted in the present form.